# Construction of PIK3C3 Transgenic Pig and Its Pathogenesis of Liver Damage

**DOI:** 10.3390/life12050630

**Published:** 2022-04-24

**Authors:** Jing Wang, Sami Ullah Khan, Pan Cao, Xi Chen, Fengchong Wang, Di Zou, Honghui Li, Heng Zhao, Kaixiang Xu, Deling Jiao, Chang Yang, Feiyan Zhu, Yaxuan Zhang, Yanhua Su, Wenmin Cheng, Baoyu Jia, Yubo Qing, Muhammad Ameen Jamal, Hong-Ye Zhao, Hong-Jiang Wei

**Affiliations:** 1Key Laboratory for Porcine Gene Editing and Xenotransplantation in Yunnan Province, Kunming 650201, China; kmwangjing@163.com (J.W.); samiullahakbar4@gmail.com (S.U.K.); caopan_2015@163.com (P.C.); 15215066081@163.com (X.C.); wfc8758369@163.com (F.W.); yunnanzoudi@163.com (D.Z.); honghui8300@aliyun.com (H.L.); hengzhao2014@126.com (H.Z.); tsljmuch@163.com (K.X.); jiaodeling@163.com (D.J.); theurgytheurgy@163.com (C.Y.); yunnanfeiyan@163.com (F.Z.); zyx1zyx111@163.com (Y.Z.); 2013003@ynau.edu.cn (Y.S.); cheng_8097@163.com (W.C.); jiabaoyu2009@163.com (B.J.); qingyubo20@163.com (Y.Q.); drameen007@gmail.com (M.A.J.); 2Xenotransplantation Research Engineering Center in Yunnan Province, Yunnan Agricultural University, Kunming 650201, China; 3Faculty of Animal Science and Technology, Yunnan Agricultural University, Kunming 650201, China; 4State Key Laboratory for Conservation and Utilization of Bio-Resources in Yunnan, Yunnan Agricultural University, Kunming 650201, China; 5College of Veterinary Medicine, Yunnan Agricultural University, Kunming 650201, China

**Keywords:** PIK3C3, transgenic pigs, liver damage, autophagy

## Abstract

As a member of the PIKs family, PIK3C3 participates in autophagy and plays a central role in liver function. Several studies demonstrated that the complete suppression of PIK3C3 in mammals can cause hepatomegaly and hepatosteatosis. However, the function of PIK3C3 overexpression on the liver and other organs is still unknown. In this study, we successfully generated PIK3C3 transgenic pigs through somatic cell nuclear transfer (SCNT) by designing a specific vector for the overexpression of PIK3C3. Plasmid identification was performed through enzyme digestion and transfected into the fetal fibroblasts derived from *Diannan* miniature pigs. After 2 weeks of culturing, six positive colonies obtained from a total of 14 cell colonies were identified through PCR. One positive cell line was selected as the donor cell line for SCNT for the construction of PIK3C3transgenic pigs. Thirty single blastocysts were collected and identified as PIK3C3 transgenic-positive blastocysts. Two surrogates became pregnant after transferring the reconstructed embryos into four surrogates. Fetal fibroblasts of PIK3C3-positive fetuses identified through PCR were used as donor cells for SCNT to generate PIK3C3 transgenic pigs. To further explore the function of PIK3C3 overexpression, genotyping and phenotyping of the fetuses and piglets obtained were performed by PCR, immunohistochemical, HE, and apoptosis staining. The results showed that inflammatory infiltration and vacuolar formation in hepatocytes and apoptotic cells, and the mRNA expression of NF-κB, TGF-β1, TLR4, TNF-α, and IL-6 significantly increased in the livers of PIK3C3 transgenic pigs when compared with wild-type (WT) pigs. Immunofluorescence staining showed that LC3B and LAMP-1-positive cells increased in the livers of PIK3C3 transgenic pigs. In the EBSS-induced autophagy of the porcine fibroblast cells (PFCs), the accumulated LC3II protein was cleared faster in PIK3C3 transgenic (PFCs) thanWT (PFCs). In conclusion, PIK3C3 overexpression promoted autophagy in the liver and associated molecular mechanisms related to the activation of ULK1, AMBR1, DRAM1, and MTOR, causing liver damage in pigs. Therefore, the construction of PIK3C3 transgenic pigs may provide a new experimental animal resource for liver diseases.

## 1. Introduction

Liver disease accounts for about 2 million deaths per year around the globe—one million due to cirrhosis complications and one million due to viral hepatitis and hepatocellular carcinoma. Liver damage mainly includes chemical and drug-induced liver damage, autoimmune liver damage, and alcoholic liver damage [1,2,3,4]. Its clinical manifestations are liver necrosis, fatty liver, liver fibrosis, liver cirrhosis, and liver cancer [5,6,7,8]. At present, the treatment of liver damage is still a serious global issue. Studying the genetic and molecular basis of liver disease is critical for understanding the pathogenesis of liver disease and developing new therapeutic modalities.

Phosphatidylinositol 3-kinase (PI3K) is a member of the growth factor family, and it plays an important regulatory role in a variety of biological processes, such as growth, proliferation, survival, migration, and metabolism [9,10]. PI3K is divided into three subtypes, namely, class I PI3K, class II PI3K, and class III PI3K. Class III PI3K(PIK3C3) is a key factor involved in the autophagic process, which regulates endocytosis and autophagosome formation. Studies have shown that the activation of PIK3C3 plays an important role in autophagy [11,12]. PIK3C3-mediated prolonged activation of autophagy plays a critical role in the transition of cardiac hypertrophy in heat shock protein 27 transgenic mice [13]. PIK3C3-null mice died between E7.5 and E8.5 of embryogenesis, which may have been related to a reduction in the mTOR signaling pathway [14].

PIK3C3 is essentially used in the process of autophagy and in normal liver function [12,15]. Mice with PIK3C3-deficient livers developed hepatomegaly and hepatosteatosis, and the mechanism may be related to the inhibition of autophagy and mTOR activation [12,16,17]. PIK3C3 inactivation limits the substrate availability of mitochondrial respiration and reduces gluconeogenesis in the mouse liver [18]. Similarly, the genetic deletion of the PIK3C3 gene in the liver demonstrates the central role of PIK3C3 in controlling vesicular trafficking and autophagy in tissue-specific homeostasis and functions [12], resulting in profound cellular and other organ damage. Multiple studies have confirmed the essential role of PIK3C3 in the maintenance of liver cancer stem cells. However poor survival rates were observed in patients with high expression of PIK3C3 in hepatocellular carcinoma [19]. Further, the autophagy level was decreased in hepatocellular carcinoma cells treated with a PIK3C3 inhibitor or small interfering RNAs (siRNA), which confirms PIK3C3 as an essential factor involved in the autophagy process [19]. In another study using an MCF-7 breast cancer cell model, PIK3C3 was observed to stimulate ERK pathway-related tumor progression through the protein kinase-mediated activation of p62. PIK3C3 induces the C- δ (PKC-δ-)-mediated dependent phosphorylation of p62, which leads to positive feedback on the Nrf2-dependent transcription of oncogenes and activation of the ERK pathway. The overexpression of PIK3C3 promotes the colony formation of MCF-7 cells in vitro and tumor growth in vivo [20]. These studies indicated that either liver-specific deletion or over-expression of PIK3C3 resulting in abnormal autophagy is at least partially responsible for either hepatomegaly or abnormal autophagy [12,14,21]. Therefore, the overexpression of PIK3C3 may lead to liver damage by the prolonged activation of autophagy. However, to date, there have been no available reports describing the construction and phenotype of PIK3C3-overexpressing animals.

*Diannan* miniature pig, a famous local variety, has unique characteristics, including early sexual maturity, small size, and being easy to handle and manage. Further, it has a higher birth rate and lower full-grown body weight than the Large White pig [22]. Moreover, a higher cloning efficiency in terms of its high litter size was found for *Diannan* miniature pigs when compared with the 19 different donor cell types from other pig breeds [22,23]. Thus, these pigs are considered as an ideal model for a variety of biomedical-related research.In the present study, we generated PIK3C3 transgenic *Diannan* miniature pigs via gene overexpression and SCNT technology to investigate the role and underlying mechanisms of PIK3C3 overexpression in liver function. We performed systematic phenotypic analysis of the PIK3C3 transgenic piglets obtained after cloning through somatic cells. These genetically engineered *Diannan* miniature pigs may provide a new experimental animal resource for the functional study of PIK3C3 in liver diseases.

## 2. Materials and Methods

### 2.1. Animals and Chemicals

The animals used in this study were regularly maintained in the Laboratory Animal Centre of Yunnan Agricultural University. All animal experiments were performed with the approval of the Animal Care and Use Committee of Yunnan Agricultural University. All chemicals were obtained from Sigma (St. Louis, MO, USA) unless otherwise stated.

### 2.2. Vector Construction

According to the PIK3C3 gene sequence (GenBank No: AY823302.1), a 2664 bp fragment of the gene-coding region was artificially synthesized and cloned into the pUC57 plasmid, and then *BamHI* and *Xho I* restriction sites were added to the plasmid to obtain the pUC57-PIK3C3 plasmid. Then, pUC57-PIK3C3 and pIRES2-AcGFP1 were simultaneously cut with *BamHI* and *Xho I* enzymes, and the corresponding fragments were recovered and ligated to obtain the pPIK-IRES2-AcGFP1 plasmid. Using PCR amplification, a protective base fragment (EN)with a length of 1450 bp was obtained, and its 5′ and 3′ ends retained a homologous sequence of 20 bp shared with the linearized vector described above. A *Sal I* restriction site was added to the 5′ end to linearize the pPIK-IRES2-AcGFP1 vector, which was then mixed with EN, and ligated and transformed with the PEASY-uni Seamless Cloning and Assembly Kit to obtain the pPIK-IRES2-AcGFP1-EN vector. Then, the plasmids were verified by enzyme digestion. The sequences information of pUC57-PIK3C3 and pPIK-IRES2-AcGFP1-ENplasmid was shown in Appendix A.

### 2.3. Cell Transfection, Selection, and Identification

*Diannan* miniature porcine fetal fibroblasts were thawed and cultured in DMEM containing 10% FBS. Approximately 7 × 10^5^ cells suspended in electro-transfection buffer were mixed with 10.5 μg of the PIK3C3 transgenic plasmids in a final sample volume of 700 μL. The cell suspension was loaded into a 4 mm-gap cuvette and subjected to an electrical treatment of one pulse at 250 V for 20 ms (Bio-Rad Gene Pulser Xcell, Hercules, CA, USA). The cells were then seeded in 5 mL of DMEM containing 10% FBS in a T-25 culture flask and incubated for 48 h at 38 °C. The cells were then detached by 0.25% trypsin, seeded into 100 mm-diameter dishes with 3000 cells/dish, and selected by G418 (600 μg/mL) after 24 h of incubation. After 12–14 days, the colonies were picked and seeded in 48-well plates. The expression of PIK3C3 was tested by PCR (Primers F: TTGGGGACAGGCACCTGGATAATC; R: CTGGGTGGACAGGTAGTGGTTATC).

### 2.4. SCNT and Generation of PIK3C3 Transgenic Fetuses and Piglets

Oocyte collection, IVM culturing, and SCNT were performed as described in our previous studies [24,25]. Donor cells from a PIK3C3-positive fibroblast cell line were inserted into the perivitelline space of an enucleated oocyte. The recombinant embryos were surgically transferred into the oviducts of the recipients. Pregnancy was confirmed approximately 23 days after surgical transfer using an ultrasound scanner (HS-101 V, Honda Electronics Co., Ltd., Yamazuka, Japan).

### 2.5. Identification of PIK3C3 Transgenic Piglets

After the birth of the cloned piglets, the ear tissues were collected, and DNA was extracted using a tissue DNA kit (Takara, Japan). The PIK3C3-modified piglets and WT piglets were identified by PCR.

### 2.6. Hematoxylin-Eosin Staining

Pig tissue was fixed with 4% paraformaldehyde for 48 h and subjected to dehydration, wax-dipping, and embedding treatment. The tissue was sliced into 5 µm slices, and the slices were incubated in a 60 °C incubator for 15 min. The slices were then deparaffinized in xylene and hydrated via gradient alcohol. The slices were rinsed for 3 min. Then, the slices were soaked in hematoxylin for 4 min, rinsed for 4 min, soaked in 1% ethanol hydrochloride for 3 s, and rinsed in water for 3 min. The slices were subsequently stained in Ehong staining solution for 3 min and dehydrated in an alcohol gradient. Finally, the slices were soaked in dimethylbenzene for 10 min, sealed with neutral resin, and observed under a microscope. The mean integrated optical intensity of the HE-staining images was measured using the Image-Pro plus application.

### 2.7. Liver Cell Apoptosis Detection

Hepatocyte apoptosis was detected using a BrightRed Apoptosis Detection Kit (Vazyme, A113, Nanjing, China). The liver tissue slices were deparaffinized with xylene and hydrated with an alcohol gradient. Liver tissue apoptosis staining was performed according to the manufacturer’s instructions. Then, the sections were observed under a fluorescence microscope (Olympus BX53, Tokyo, Japan). The mean integrated optical intensity of the apoptosis-staining images was measured using the Image-Pro plus application.

### 2.8. Immunofluorescence Staining

WT and PIK3C3 transgenic PFCs were seeded onto 6-well culture slides and treated with EBSS for 2 h, fixed with formaldehyde for 10 min, and then blocked with a buffer containing 5.0% BSA and 0.5% Triton X-100 for 30 min. The liver tissue slices were dewaxed with xylene and soaked with an alcohol gradient, and the tissue slices were then blocked with a buffer containing 5.0% BSA and 0.5% Triton X-100 for 30 min. Next, the cells and tissues were incubated with a primary antibody against LC3 (1:200), LAMP-1 (1:200), at 4 °C overnight. Then, the cells were incubated for 1 h with a secondary FITC-conjugated antibody (1:400) and CY3-conjugated antibody (1:400) to visualize the binding sites of the primary antibody with microscopy (Olympus BX53, Tokyo, Japan). The mean integrated optical intensity of the immunofluorescence staining images was measured using the Image-Pro plus application.

### 2.9. Immunohistochemical Staining

WT and PIK3C3 transgenic tissues were dewaxed with xylene and soaked with an alcohol gradient. Next, the expression of PIK3C3 (1:200, Rabbit, Cell Signaling) in the heart, liver, spleen, lung, and kidney tissues was tested by immunohistochemical staining. The immunohistochemical staining was performed as previously described [26]. The mean integrated optical intensity of the immunohistochemical staining images was measured using the Image-Pro plus application.

### 2.10. Transmission Electron Microscopy

WT and PIK3C3 transgenic PFCs were treated with EBSS for 0 h, 1 h, and 2 h, and were fixed overnight at 4 °C using 2.5% glutaraldehyde in PBS. Afterward, the samples were postfixed with 1% OsO_4_ for 2 h at 4 °C, followed by serial ethanol dehydration and embedding in Epon 812 resin. Serial sections with uniform thicknesses of approximately 60 nm were produced using a Leica UC7 ultramicrotome. After staining with 2% uranyl acetate and lead citrate, ultrathin sections were examined using a transmission electron microscope (JEM 1400 plus, JEOL, Tokyo, Japan). The total number of autophagosomes and autolysosomes in each image was counted to quantify autophagy in each group.

### 2.11. q-PCR

The total RNA was extracted via the TRIzol method. Two micrograms of the total RNA were reverse-transcribed with the Prime Script^TM^ RT reagent kit with gDNA Eraser (TAKARA, Ca: RR047A). The relative gene expression levels were calculated with the following formula: X = 2^−ΔΔCt^. Part of the specific primers used are shown in Table 1, and the other primers are quoted from our previous study [27].

### 2.12. Western Blotting

The tissue and cells were lysed in total protein extraction buffer (Code^#^:DE101, Lot^#^:10809) with protease inhibitors at 4 °C. After centrifugation at 14,000× *g* for 15 min at 4 °C, the supernatant was collected. The protein concentration was determined by the BCA method. Then, 5 × loading buffer was added to the lysate according to the ratio of 1:4 (*v*/*v*). The protein sample was denatured in 95 °C water for 5 min. The protein sample was then stored at −80 °C. The protein was separated by SDS–Page. After electrophoresis, the protein was transferred to the PVDF membrane (200 mA, 2 h). The PVDF membrane was blocked in 5% skimmed milk powder solution for 2 h at room temperature. Then. the membrane was rinsed and incubated overnight in the first antibody at 4 °C. After rinsing with 1 × TBST, the PVDF membrane was incubated in the second antibody for 2 h at room temperature. The membrane was developed in the ECL detection system (Easysee Western Blot Kit, Transgen, Beijing, China), and the gray value was analyzed by Image J software.

### 2.13. Statistical Analysis

All values were expressed as the mean ± SD based on 3 independent experiments and statistically analyzed by using SPASS 17.0 software. The PIK3C3 mRNA level in different organs, inflammatory factor mRNA level in liver tissue, ATG and autophagy signaling pathway genes mRNA level in liver and spleen, and the PIK3C3 protein level in WT and PIK3C3 transgenic PFCs were statistically analyzed by independent-sample *t*-tests. The mRNA levels of ATG and autophagy signaling pathway genes in WT and PIK3C3 transgenic PFCs were statistically analyzed by one-way ANOVA coupled with LSD and Dunnett’s T3 post hoc tests. The statistical significance was set as *^, #^ *p* <0.05 and **^, ##^
*p* < 0.01.

## 3. Results

### 3.1. Construction of PIK3C3 Transgenic Plasmid and PIK3C3 Transgenic Fetuses and Piglets

We generated a specific vector for the overexpression of PIK3C3 (Figure 1A), which was driven by a CMV promoter. The plasmid was identified via enzyme digestion (Figure 1B) and transfected in *Diannan* miniature pig fetal fibroblast cells via electroporation. After 2 weeks of culturing, six positive colonies were obtained from a total of 14 cell colonies by PCR identification (Figure 1C). Then, one positive cell colony was used as a donor cell for SCNT. Thirty single blastocysts were collected and identified as PIK3C3 transgenic-positive blastocysts (Figure 1D).

A total of 994 cloned embryos were transferred to four surrogates, and two of them became pregnant. Five morphologically normal fetuses were obtained from the two recipients after 33 days, and their fetal fibroblasts were isolated and cultured (Table 2). As shown by PCR identification, all five fetuses were PIK3C3-positive (Figure 1E). Fetal fibroblasts of the PIK3C3-positive fetuses were used as donor cells for the second round of SCNT in the generation of PIK3C3-transgenic pigs. According to our previous work, using fibroblast cells derived from fetuses obtained by the first round of cloning can retain the cellular vitality and purify the genotype of gene editing animals [24]. Hence, we performed the re-cloning process to obtain PIK3C3-transgenic piglets.

In the second round cloning, a total of 3262 reconstructed embryos were transferred to 10 recipients, 6 recipients were pregnant and 4 surviving cloned piglets were obtained from 2 recipients after 121 days (Table 3). The 4 PIK3C3 transgenic piglets were identified by PCR (Figure 1F). Among the 4 PIK3C3 transgenic pigs, one died immediately after birth, and the remaining three died at 10 days, 20 days, and 62 days, respectively (Table 4).

### 3.2. PIK3C3 Overexpression Identification Detection of PIK3C3 Transgenic Piglets

The mRNA expression levels in the various tissues of the PIK3C3 transgenic and WT piglets were detected, and the results showed that PIK3C3 mRNA expression in the heart, liver, spleen, lung, kidney, and muscle tissues was significantly increased compared to that in the WT piglets, which confirmed the overexpression of PIK3C3 (Figure 2A). The PIK3C3 protein expression in the heart, liver, spleen, lung, and kidney was tested by the immunochemical staining of four PIK3C3 transgenic pigs, and the results showed that the PIK3C3 protein levels were significantly increased compared with those of the WT piglets, especially in the liver (Figure 2B,C).

### 3.3. PIK3C3 Overexpression Induces Inflammatory Infiltration, Cell Death, and Autophagy in Liver

The pathological examination by HE staining showed that the morphology of the WT pig liver tissue was normal, the structure of the liver lobule was complete, the arrangement of liver cells was regular, and no hepatocyte damage, such as steatosis, necrosis, or inflammatory cell infiltration, was found. Inflammatory cell infiltration was found in the liver tissue of the PIK3C3 transgenic pigs, liver cells were arranged in a disorderly manner, and vacuole formation was observed in the cytoplasm (Figure 3A,C). To further study the reasons for the pathological histological changes involved in the liver of the PIK3C3 transgenic piglets, we assessed liver cell apoptosis. The results showed that apoptotic liver cells increased significantly in the PIK3C3 transgenic pigs (Figure 3B,C). The mRNA expression of inflammatory factorsTGF-β1, TNF-α, and IL-6 increased significantly in the livers of the PIK3C3 transgenic pigs. We also observed increases in NF-κB andTLR-4, which were key factors of the NF-κB/TLR-4 signaling pathway (Figure 3D). The pathological changes, apoptosis of liver cells, increase in inflammatory factors, and NF-κB/TLR-4 signaling pathway indicated that the overexpression of PIK3C3 induced liver damage in the PIK3C3 transgenic pigs.

PIK3C3 is a key factor in autophagy, which controls intracellular vesicular trafficking, and the overactivation of autophagy leads to cell apoptosis [28]. In order to explore the underlying mechanism of liver damage caused by PIK3C3 overexpression, we observed autophagic activation in the PIK3C3 transgenic pigs in vivo. Lysosomes degrade the contents wrapped in autophagy to achieve cell homeostasis and the renewal of organelles. Therefore, lysosomes are essential for the autophagy process. LAMP1 is the main lysosomal membrane glycoprotein, accounting for about 50% of the lysosomal membrane protein. Currently, LAMP1, as a lysosome marker, can be used to detect the formation of autophagic lysosomes. To verify whether PIK3C3 overexpression promotes the process of autophagy, the LC3B and LAMP-1 protein levels in the liver tissue were measured by immunofluorescence staining. Compared with those of the WT pigs, the LC3 and LAMP-1-positive cells were increased in the PIK3C3 transgenic pigs (Figure 4A,B). These results indicated that the overexpression of PIK3C3 triggered autophagy in the livers of the PIK3C3 transgenic pigs. We also found the activation of ATG and autophagy signaling pathway genes. The mRNA expression level of ULK1, AMBR1, DRAM1, and MTOR genes was increased significantly in transgenic pigs, while ATG4B and RAB1A mRNA expression level in the Liver were similar between transgenic pigs and control (Figure 4C). Compared with the WT group, the mRNA level of ULK1, AMBR1, DRAM1, and MTOR increased significantly, but those of ATG4B and RAB1A did not change in the liver (Figure 4C). As PIK3C3 is also expressed in immune cells, we tested the ATG and autophagy-related signaling pathway genes’ mRNA in the spleen. The results showed that the mRNA levels of ULK1, AMBR1, DRAM1, FOXO1, and MTOR increased significantly, but those of ATG4B and RAB1A did not change in the spleen (Figure 4D).

### 3.4. PIK3C3 Promotes Autophagy in PIK3C3 Transgenic PFCs In Vitro

Here, we also verified the effects of PIK3C3 overexpression on autophagy in vitro. PFCs were isolated from the ear tissues of the obtained PIK3C3 transgenic piglets. The western blotting results showed that the PIK3C3 protein level increased significantly in the PIK3C3 transgenic PFCs (Figure 5A,B and Appendix A). Autophagy was induced by EBSS treatment (0 h, 1 h, and 2 h) in the WT and PIK3C3 transgenic PFCs. In order to prove the occurrence of autophagy, the autophagosomes in WT and PIK3C3 transgenic PFCS were detected by transmission electron microscopy (Figure 5C). After 2 h of EBSS treatment, the total number of autophagosomes and autophagolysosomes increased in WT PFCs, but decreased in PIK3C3 transgenic PFCs (Figure 5D,E).

As LC3B is the marker protein of autophagy, we also tested the expression of LC3B in WT and PIK3C3 transgenic PFCs by immunofluorescence staining (Figure 6A). The results showed that the fluorescence intensity of the LC3B protein in PIK3C3 transgenic PFCs was stronger than that of WT PFCs before EBSS treatment. In WT PFCs, the LC3 protein intensity increased with the EBSS treatment time. In PIK3C3 transgenic PFCs, the protein fluorescence intensity of LC3B gradually decreased with the EBSS treatment time (Figure 6B,C). The accumulation of LC3B might have already occurred before EBSS treatment for 1 h. The accumulated LC3B began to be degraded at 1 h. These results indicate that PIK3C3 promoted the occurrence of autophagy.

In order to further explore the molecular mechanism that PIK3C3 overexpression promotes autophagy, the ATG and autophagy signaling pathway-related gene mRNA expression levels were detected in WT and PIK3C3 transgenic PFCs (Figure 7). Compared with the control group, the mRNA levels of ULK1, ATG2B, AMBR1, RAB1B, and MTOR in WT and PIK3C3 transgenic PFCs were significantly increased after EBSS treatment for 1 h and 2 h, and the fold-changes of mRNA level in PIK3C3 transgenic PFCs were higher than those of WT PFCs. In WT PFCs, compared with the 1 h group, the expression levels of ULK1, ATG2B, RAB1B, and MTOR increased further in the 2 h group. However, in PIK3C3 transgenic PFCs, compared with the 1 h group, the above four factors decreased in the 2 h group. This indicates that autophagy occurred earlier in PIK3C3 transgenic PFCs, and autophagy began to weaken at 2 h.

Compared with the 0 h group, the mRNA level of ATG14 did not change after EBSS treatment for 1 h in WT PFCs. However, the mRNA level of ATG14 in PIK3C3 transgenic PFCs was significantly increased after EBSS treatment for 1 h (Figure 7A,C). Compared with the control group, the ATG4A mRNA level did not change after EBSS treatment in WT and PIK3C3 transgenic PFCs (Figure 7A–D). Compared with the 0 h group, the mRNA levels of ATG7, RAB1A, and FOXO1 did not change after EBSS treatment in WT PFCs. In PIK3C3 transgenic PFCs, the expression levels of ATG7 and FOXO1 mRNA increased, while that of RAB1A did not change after EBSS treatment for 1 h. The mRNA levels of ATG7 and FOXO1 decreased at 2 h (Figure 7A–D). In PIK3C3 transgenic PFCs, after 1 h of EBSS treatment, the expression level of RAB1A did not change, but decreased after 2 h of EBSS treatment (Figure 7B,D). Compared with the 0 h group, the expression level of DRAM2 mRNA decreased significantly for the 2 h group in WT PFCs, but it was increased significantly in PIK3C3 transgenic PFCs (Figure 7B,D). This indicates that PIK3C3 overexpression promotes autophagy in PFCs, and PIK3C3 may promote autophagy by the activation of ULK1, ATG13, ATG2B, ATG14, AMBR1, ATG7, RAB1B, DRAM2, FOXO1, and MTOR genes.

## 4. Discussion

Autophagy plays a critical role in the pathogenesis of diverse diseases, such as neuronal degeneration, aging, and cancer. Here, we report that the overexpression of PIK3C3 induces liver damage by enhancing autophagy in the liver.

In this study, we successfully obtained PIK3C3 transgenic pigs by SCNT. In the second round of SCNT, a total of 3262 reconstructed embryos were transferred to 10 recipients; six recipients became pregnant, out of which four piglets were obtained from two recipients. Four pregnant recipients did not produce piglets. In our previous study, the pregnancy rate of wild-type pig-cloned embryos in recipients was about 60%, which is consistent with our current data. However, the delivery rate of PIK3C3 transgenic pigs was only 20% as compared to that of wild-type cloned pigs (60%) [25]. Therefore, we speculate that the overexpression of PIK3C3 may have impacts on embryonic development. However, there are many factors that affect productivity. In this study, we just focused on the generation of PIK3C3 transgenic pigs and we plan to investigate the developmental defects in our next studies.

The PIK3C3 mRNA expression level of the obtained PIK3C3 transgenic pigs varied greatly among individuals (Figure 2A). The exact reason for the variation in expression is unknown, but it could be inferred that age differences (1–62 days) within transgenic pigs might have contributed to variations in expressions. In an epigenetic study, it was shown that the expression of hepatic glucose metabolism-related genes varied at different ages [29]. In addition, epigenetic modifications might have affected the gene expression profiles between cloned individuals, as it has been reported that the gene transcription patterns differ significantly among cloned embryos, especially among cloned embryos that fail to develop a normal phenotype [30,31,32]. In addition, there are also other reasons that may lead to the variation of the PIK3C3 gene level. PIK3C3 transgenic pigs were constructed using random insertion and CMV as the promoter. The CMV promoter is prone to silencing [33] and the insertion site of transgenes might also lead to unknown position effects that usually happen in random integrations [34].

PIK3C3 transgenic pigs showed symptoms of liver damage with regard to pathology, inflammation, and hepatocyte apoptosis (Figure 3). Studies have reported the formation of vacuoles and hepatocyte apoptosis in liver damage, which is consistent with our research [35]. We analyzed the causes of liver inflammation in PIK3C3 transgenic pigs. Among the PI3K family, PIK3C3 is highly expressed and well-characterized in immune cells [36]. Studies have reported that canonical autophagy dependent on PIK3C3 is required for naive T-cell homeostasis [37]. The overexpression of PIK3C3 may lead to excessive activation of T cells. PIK3C3 has been suggested to play an important role in autoimmune diseases [38]. In autoimmune diseases, T cells are in a state of excessive activation and hyperproliferation, and the hyperproliferation of T cells leads to a cascade of inflammatory mediators released through effector cytokines [39]. In the PIK3C3 transgenic pigs, we also observed an increase in the TLR4, NF-κB, IL-6, TNF-α, and TGF-β1 mRNA levels in the liver tissue (Figure 3D). TLR4, an immune receptor with a crucial role in pathogen recognition and the activation of innate immunity, was increased [40]. When TLR4 binds to the corresponding ligand, it can further activate the NF-κB signaling pathway and then promote the expression of inflammatory cytokines, such as IL-6, TNF-α, and TGF-β1 [41,42,43,44]. The activation of the TLR4-NF-κB signaling pathway has been reported to be related to the occurrence of autoimmune liver damage [45]. IL-6, TNF-α, and TGF-β1 are important inflammatory factors in the inflammatory response of the liver and are involved in a variety of liver diseases [46,47]. Studies have also reported increases in IL-6, TNF-α, and TGF-β1 in liver damage, which is consistent with our results [48]. Combined with liver pathological damage and the increase in inflammatory factors, the overexpression of PIK3C3 may affect T-cell homeostasis and activate the TLR4-NF-κB signaling pathway, leading to liver damage. PIK3C3 has been suggested to play an important role in autoimmune diseases [49]. However, because of the limitation of samples, properly assessing the immune cells in terms of investigating the relationship between PIK3C3 and autoimmune disease in PIK3C3 overexpressed pigs still needs further study.

Autophagy is a process in which cytoplasmic components and entire organelles are targeted by lysosomes for degradation. Under normal conditions, autophagy plays an important role in the turnover of organelles at low basal levels [50]. However, the prolonged activation of autophagy causes abnormalities and dysfunction of intracellular compartments [51,52,53]. In the present study, we found liver cell apoptosis (Figure 3B) and prolonged activation of autophagy (Figure 4) in the PIK3C3 transgenic pigs. In vitro experiments indicated that PIK3C3 overexpression promotes the autophagy response of PFCs, which is manifested in the acceleration of autophagy lysosome formation and the degradation of LC3B (Figure 5 and Figure 6). Studies have reported that PIK3C3-mediated prolonged activation of autophagy leads to cardiac hypertrophy in HSP-27 transgenic mice [12,13]. The inhibition of PIK3C3-dependent autophagy could prevent neuroblastoma cell apoptosis and necrosis through oxidative stress [54]. The overactivation of the class-III PI3K/Beclin 1-dependent autophagic signals induced primary rat astrocyte apoptosis [55]. These reports indicate that the prolonged activation of autophagy leads to cell apoptosis, which is consistent with our results. The hyperinduction of autophagy and consequent excessive lysosomal degradation of cell constituents may lead to so-called “autophagic cell death” [28]. In our study, the PIK3C3 transgenic pigs had increased hepatocyte apoptosis and autophagy (Figure 3 and Figure 4), which was consistent with this report. These results indicate that liver damage in PIK3C3 transgenic pigs may be related to the excessive activation of autophagy.

In the liver tissue of PIK3C3 transgenic pigs, the protein levels of LC3 and LAMP-1 both increased compared with those of the WT pigs. Similar findings have been already reported in transgenic mice, in which the expression level of LC3B also increased [13]. However, in the cell experiment, because the cells were very sensitive to starvation stress, when autophagy in the cells was induced by EBSS treatment, the autophagy level was very strong. In the EBSS-treated PIK3C3 transgenic cells, LC3B accumulated rapidly and degraded rapidly. The accumulated LC3B was degraded after 2 h of EBSS treatment. Therefore, we did not observe the process of LC3B accumulation in PIK3C3 transgenic cells. We measured the intensity of the LC3B immunofluorescence images of PIK3C3 transgenic cells, the results also showed that LC3B decreased after 2 h of EBSS treatment (Figure 6C). The degradation of ATG-related protein was not so fast; therefore, we observed that ATG increased and LC3B decreased after 2 h of EBSS treatment. In addition, compared with the 1 h group, the mRNA level of ATG began to decrease when EBSS was treated for 2 h (Figure 7C,D).

In order to study the mechanism of PIK3C3 leading to autophagy activation, we tested the ATG gene and autophagy signaling pathway-related genes and found that the mRNA levels of ULK1, AMBR1, DRAM1, and MTOR increased in the liver and spleen tissues (Figure 4C,D). Additionally, in the starvation-induced autophagy experiment, PIK3C3 increased the fold-changes of ULK1, AMBR1, and MTOR mRNA in PIK3C3 transgenic PFCs compared with WT PFCs (Figure 7). In the process of autophagy, the induction of autophagosome formation requires the activation of the ULK1/ATG1 kinase complex, and it mainly integrates upstream molecular signals through two key kinases, MTOR and PRKA/AMPK [56]. Proautophagy factor AMBAR is the key link between ULK1 and BECLIN1 complex 1, and it is one of the few known functional targets of MTOR [57]. Studies reported that the activation of AMPK and ULK1 was associated with increased BECN1 (S93 and S14) and PIK3C3/VPS34 (S164) phosphorylation, as well as increased total ATG14 and PIK3C3. EtOH-induced phosphorylation and dephosphorylation of ULK1 at S555 and S757 alters the interaction between AMPK and ULK1, and increases the association of ULK1 with BECN1, ATG14, and PIK3C3 [58]. However, further studies are required to properly detect the effect of PIK3C3 overexpression on p-ULK1.

Previous studies have shown that DRAM1 induces autophagy by inducing autophagic lysosome aggregation. DRAM1 acts as a downstream of p53 and induces p53 and p73-dependent autophagic apoptosis [59,60,61]. PIK3C3 overexpression may activate autophagy by activating ULK1, AMBR1, DRAM1, and MTOR, which may lead to liver damage. After EBSS treatment for 2 h, ATG13 and ATG14 in PIK3C3 transgenic PFCs increased significantly, but there was no significant change in WT PFCs (Figure 7). ATG13 is located on the autophagy isolation membrane and is necessary for the formation of autophagosomes. It forms a ULK1–Atg13–FIP200 complex with ULK1 and FIP200 [56]. ATG14 is an important autophagy-specific regulator of the class-III phosphatidylinositol 3-kinase complex, which promotes the membrane binding of non-proteoliposomes and enhances the semi-fusion and complete fusion of recombinant proteoliposomes [62]. PIK3C3 overexpression may activate autophagy by activating ULK1, ATG13, ATG14, AMBR1, and MTOR in PFCs.

In this study, we successfully generated PIK3C3 transgenic *Diannan* miniature pigs. In the liver, PIK3C3 overexpression triggered prolonged the activation of autophagy by the activation of ULK1, AMBR1, DRAM1, and MTOR, and then caused liver damage in the pigs. The construction of PIK3C3 transgenic pigs may provide a new experimental animal resource for PIK3C3 function studies in liver diseases.

## Figures and Tables

**Figure 1 life-12-00630-f001:**
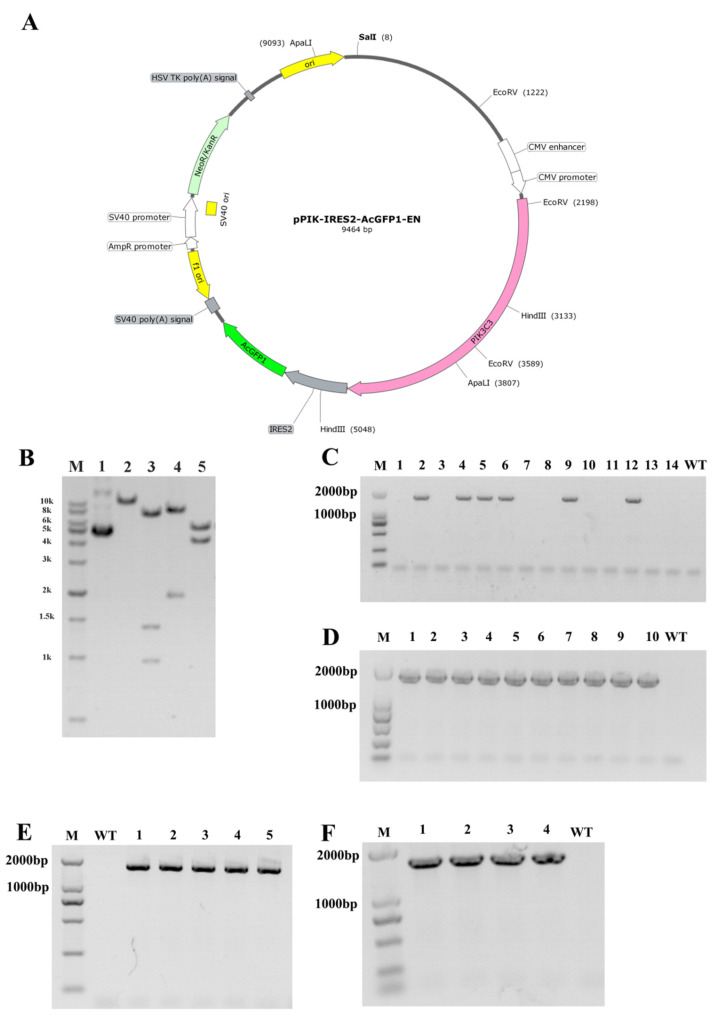
Construction of PIK3C3 transgenic plasmid and PIK3C3 transgenic piglets. (**A**) Schematic diagram of the plasmid for PIK3C3 overexpression. (**B**) Identification of the plasmids by the restricted enzyme method (M, 1kb DNA Marker; 1, Circular pPIK-IRES2-AcGFP-EN plasmid; 2–5, the pPIK-IRES2-AcGFP-EN plasmid was digested by *Sal I*, *EcoR V*, *Hind III*, and *Apal I* endonuclease respectively, and the product sizes were 9654 bp, 982 bp + 1391 bp + 7094 bp, 1913 bp + 7554 bp, and 4184 bp + 5283 bp. (**C**) Identification of the PIK3C3 transgenic single-cell colony by PCR (M, DL2000 DNA Marker; C, colony). The band size of the PIK3C3 transgenic PFCs is 1652 bp. (**D**) Identification of a part of the PIK3C3 transgenic single blastocyst (M, DL2000 DNA Marker). The band size of the PIK3C3 transgenic single blastocyst is 1652 bp.(**E**) Five PIK3C3 transgenic fetuses were identified by PCR detection (M, DNA marker DL2000). The band size of PIK3C3 transgenic fetuses is 1652 bp. (**F**) Four PIK3C3 transgenic piglets were identified by PCR detection (M, DNA marker DL2000). The band size of PIK3C3 transgenic piglets is 1652 bp.

**Figure 2 life-12-00630-f002:**
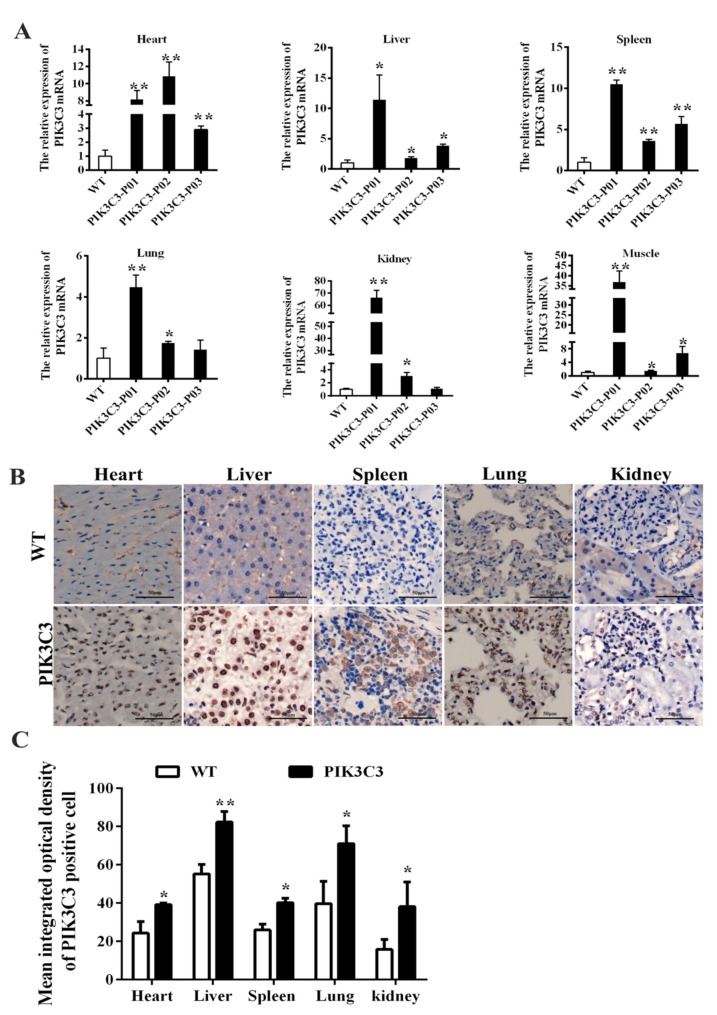
PIK3C3 overexpression identification of PIK3C3 transgenic piglets.(**A**) Relative expression levels of PIK3C3 mRNA in tissues from WT and PIK3C3 transgenic piglets. The relative expression levels of PIK3C3 mRNA in the heart, liver, spleen, lung, kidney and muscle of WT and PIK3C3 transgenic piglets were measured using q-PCR. The expression of the GAPDH gene was used to normalize the values of PIK3C3. (**B**) Immunohistochemical staining of PIK3C3 in the heart, liver, spleen, lung, and kidney from WT and PIK3C3 transgenic piglets. Scale bar = 50 μm. (**C**) Mean integrated optical density of positive cells in immunohistochemical staining of PIK3C3 in different tissues (heart, liver, spleen, lung, and kidney) from WT and PIK3C3 transgenic piglets. Scale bar = 50 μm. * *p* < 0.05 and ** *p* < 0.01 denote significant differences in PIK3C3 transgenic piglets compared to WT piglets.

**Figure 3 life-12-00630-f003:**
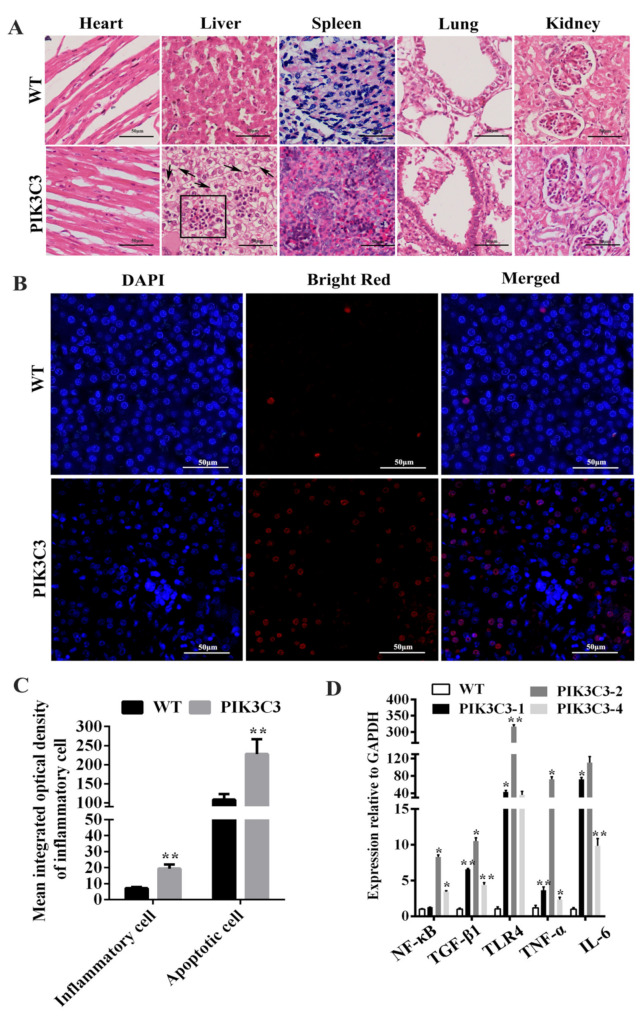
PIK3C3 overexpression promoted liver damage. (**A**) HE staining in the heart, liver, spleen, lung and kidney from WT and PIK3C3 transgenic piglets. Scale bar = 50 μm. Inflammatory cells are shown in black boxes, and liver tissue vacuoles are indicated by black arrows. (**B**) Liver cell apoptosis detection in WT and PIK3C3 transgenic piglets. The nuclei were stained blue by DAPI and the apoptotic cells were stained red by bright red. Scale bar = 50 μm. (**C**) Mean integrated optical density of inflammatory cells in HE staining and apoptotic cells in the liver tissue of WT and PIK3C3 transgenic piglets. (**D**) Relative expression levels of NF-κB, TGF-β1, TLR4, TNF-α, and IL-6 mRNA in liver tissue from WT and PIK3C3 transgenic piglets were measured using q-PCR. The expression of the GAPDH gene was used to normalize the values of NF-κB, TGF-β1, TLR4, TNF-α, and IL-6. * *p* < 0.05 and ** *p* < 0.01 denote significant differences in PIK3C3 transgenic piglets compared to WT piglets.

**Figure 4 life-12-00630-f004:**
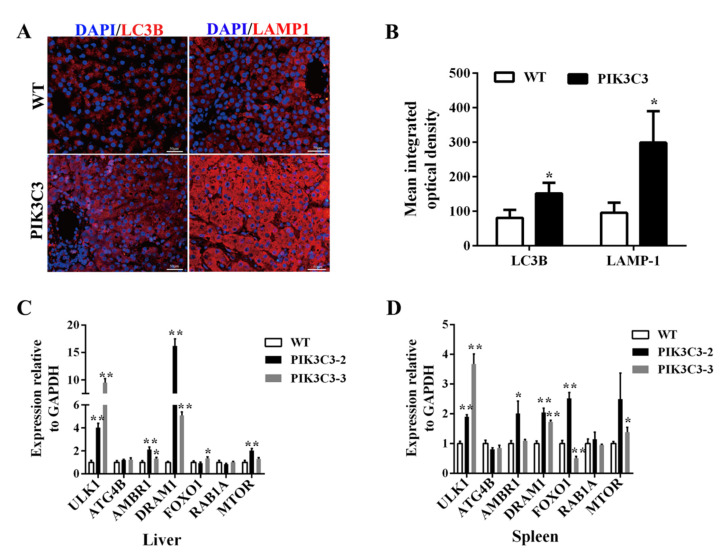
PIK3C3 overexpression promoted autophagy in liver. (**A**) Immunofluorescence staining of LC3B and LAMP-1 in the liver tissue of WT and PIK3C3 transgenic piglets. Scale bar = 50 μm. (**B**) Mean integrated optical density of the immunofluorescence staining of LC3B and LAMP-1 in the liver tissue of WT and PIK3C3 transgenic piglets. (**C**) Relative expression levels of ATG and autophagy signaling pathway genes’ mRNA in the liver tissue from WT and PIK3C3 transgenic piglets measured by using q-PCR. (**D**) Relative expression levels of ATG and autophagy signalingpathway genes’ mRNA in the spleen tissue from WT and PIK3C3 transgenic piglets measured using q-PCR. The expression of the GAPDH gene was used to normalize the values of ATG. * *p* < 0.05 and ** *p* < 0.01 denote significant differences in PIK3C3 transgenic piglets compared to WT piglets.

**Figure 5 life-12-00630-f005:**
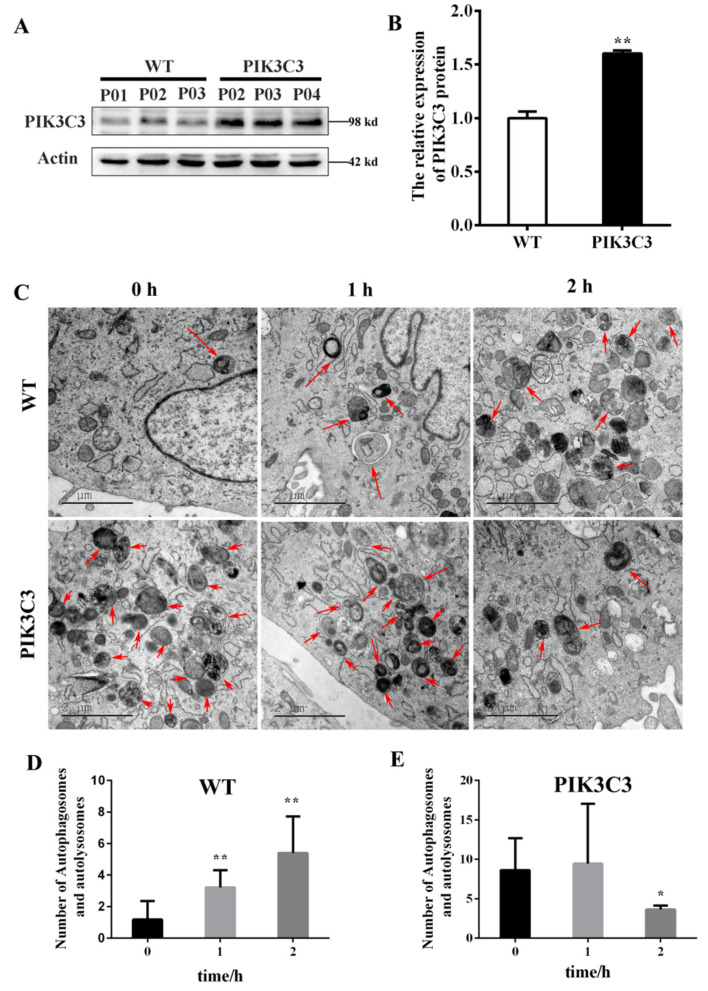
EBSS induced autophagy in WT and PIK3C3 transgenic PFCs. (**A**,**B**) Protein levels of PIK3C3 in WT and PIK3C3 transgenic PFCs measured by western blot. The expression of the β-actin was used to normalize the values of PIK3C3. * *p* < 0.05 denotes significant differences in the PIK3C3 transgenic PFCs compared to the WT PFCs. (**C**–**E**) WT and PIK3C3 transgenic PFCs treated with EBSS for 0, 1, and 2 h. The autophagosomes and autophagolysosomes were observed using a transmission electron microscope. The autophagolysosome and autophagosome are indicated by red arrows. The total number of autophagolysosomes and autophagosomes in each image was counted. Scale bar = 2 μm. * *p* < 0.05, ** *p* < 0.01 compared with the 0 h group.

**Figure 6 life-12-00630-f006:**
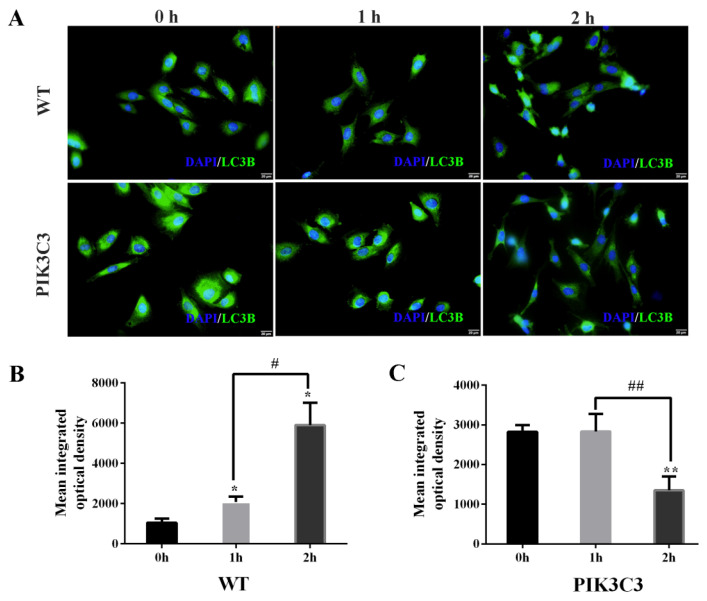
Effect of PIK3C3 overexpression on LC3B expression in PFCs. (**A**) Immunofluorescence analysis of LC3B in PFCs isolated from WT and PIK3C3 transgenic piglets. WT and PIK3C3 PFCs were treated with EBSS for 0, 1, and 2 h. Then, the expression levels of LC3B from WT and PIK3C3 transgenic PFCs were tested by immunofluorescence staining. PFCs were stained with anti-LC3B secondary antibody (green). DAPI (blue) staining indicates the nucleus. Scale bar = 20 μm. (**B**,**C**) Mean integrated optical density of the immunofluorescence staining of LC3B in WT and PIK3C3 transgenic PFCs. Compared with the 0 h group, * *p* < 0.05, ** *p* < 0.01; 1 h group compared with the 2 h group, ^#^
*p* <0.05, ^##^
*p* <0.01.

**Figure 7 life-12-00630-f007:**
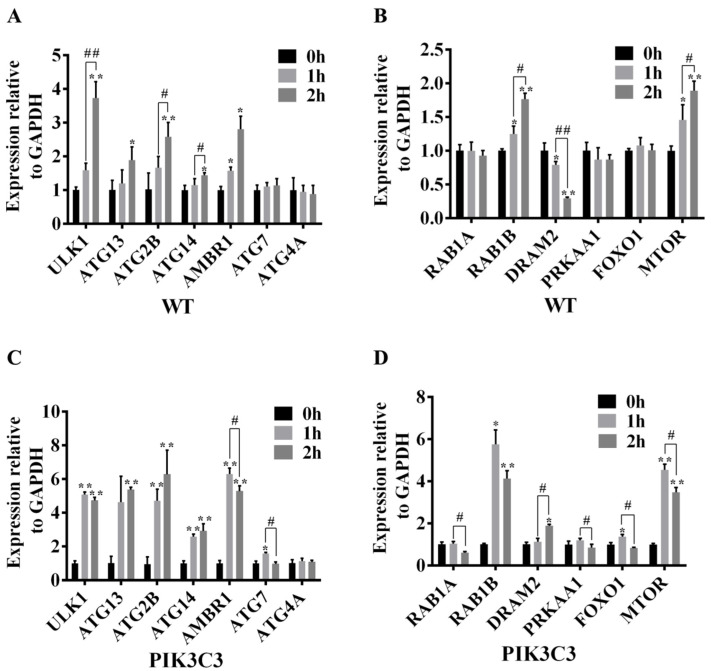
Effect of PIK3C3 overexpression on the expression levels of ATG and autophagy signaling pathway-related genes in PFCs. (**A**) mRNA levels of ULK1, ATG13, ATG2B, ATG14, AMBR1, ATG7, and ATG4A in WT PFCs. (**B**) mRNA levels of the RAB1A, RAB1B, DRAM2, PRKAA1, FOXO1, and MTOR in WT PFCs. (**C**) mRNA levels of ULK1, ATG13, ATG2B, ATG14, AMBR1, ATG7, and ATG4A in PIK3C3 transgenic PFCs. (**D**) mRNA levels of RAB1A, RAB1B, DRAM2, PRKAA1, FOXO1, and MTOR mRNA in PIK3C3 transgenic PFCs. GAPDH was used as an internal reference. Compared with the 0 h group, * *p* < 0.05, ** *p* < 0.01; 1 h group compared with the 2 h group, ^#^
*p* < 0.05, ^##^
*p* < 0.01.

**Table 1 life-12-00630-t001:** q-PCR primer sequence.

Gene Name	Forward	Reverse
IL-6	CAGGAGACCTGCTTGATGAGAA	GCCTCGACATTTCCCTTATTGC
TGF-β1	GAGCCCTGGATACCAACTACTG	TGGGTTCATGAATCCACTTCCA
TLR4	TTGAACAGTTCCGGATAGCACA	GGCTTCTAGACCACGCAAATTC
TNF-α	CAACGTTTTCCTCACTCACACC	CAGGTAGATGGGTTCGTACCAG
NF-κB	TACACTGAAGCCATTGACGTGA	CTCGTCTATTTGCTGCCTTGTG
PIK3C3	CATTTTAACGGGCTTTGAGATAGTG	ATGTAAGTTGCTTGGTTGGTGGATA

**Table 2 life-12-00630-t002:** Development of reconstructed PIK3C3 cloned embryos after transferring to recipient gilts in the first round of SCNT.

Recipients	Transferred Embryos	Pregnancy Outcomes	Days of Pregnancy (d)	Offspring (fetus)
1	270	−	−	
2	260	−	−	
3	230	+	33	2
4	234	+	33	3
Total	994	2 (50%)		5

− shows pregnancy-negative, while + shows pregnancy-positive.

**Table 3 life-12-00630-t003:** Development of reconstructed PIK3C3 cloned embryos after transferring to recipient gilts in the second round of SCNT.

Recipients	Transferred Embryos	Pregnancy Outcomes	Days of Pregnancy (d)	Delivery Outcomes	Offspring (Piglets)
1	480	−	−	−	−
2	625	−	−	−	−
3	210	+	121	+	3
4	207	+	+	−	−
5	320	−	−	−	−
6	330	+	121	+	1
7	310	+	+	−	−
8	310	+	+	−	−
9	240	+	+	−	−
10	230	−	−	−	−
Total	3262	6 (60%)		2 (20%)	4

− shows pregnancy-negative, while + shows pregnancy-positive.

**Table 4 life-12-00630-t004:** Survival time of the PIK3C3 transgenic piglets.

No. of the PIK3C3 Transgenic Piglets	Survival Time of the PIK3C3 Transgenic Piglets
P1#	10 days
P2#	62 days
P3#	20 days
P4#	10 min

## Data Availability

Not applicable.

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
