# Peer review of "Construction of PIK3C3 Transgenic Pig and Its Pathogenesis of Liver Damage"

_life, 2022, doi:10.3390/life12050630_

Round 1

Reviewer 1 Report

Summary:

The authors pointed out the urgent need to understand the underlying mechanisms for liver diseases. With the goal of developing a new animal model for liver diseases, especially focusing on autophagy caused liver damages, the authors made transgenic Diannan miniature pigs that over-express PIK3C3. Livers from PIK3C3 overexpressing pigs as well as WT pigs were examined for pathology and autophagy activity. It was concluded that these transgenic pigs may provide a new experimental animal resource for understanding the PIK3C3 function study in liver disease.

While the amount of effort spent in making the transgenic pigs and the goal of developing new animal models of human disease should be encouraged and appraised, the overall design of the experiment and the characterization of the animal model of liver disease is severely flawed. With no evidence provided in the introduction that over-expressing of PIK3C3 causes liver damages, the authors made transgenic pigs constitutively overexpress PIK3C3 globally under a CMV promoter. Considering the broad function of PIK3C3, constitutive overexpression of this gene may make the results very complex and this may have contributed to the early deaths of piglets after birth. The characterization of the liver tissue is very superficial and the results were not properly presented (see major comments below). The link between the observed liver damage and PIK3C3 over-expression mediated autophagy is not convincing. The contribution of PIK3C3 hyperactivity in the immune system to liver damage needs further evidence. More importantly, all 4 of the PIK3C3 over-expressing pigs are very short-lived, with the oldest one lived only for 30 days. This is too young to reproduce and breed, which makes it unlikely to be conveniently used as an experimental animal model by others.

Major concerns:

  1. Line 41-42: this sentence is confusing: “The accumulated LC3II protein was cleared faster in EBSS-induced autophagy of the WT and PIK3C3 transgenic porcine fibroblast cells (PFCs).”

  1. Constitutive and global over-expression of PIK3C3. Developmental defects are expected and the author should describe those defects in detail, if the transgenic pigs are to be used as a liver damage animal model.
  2. The insertion method using by the authors results random insertion(s) of the donor DNA fragment in the host genome. Insertion of transgene may cause mutations of the endogenous gene. Multiple insertions of the transgene is also highly possible. The author only used PCR to detect the presence of the transgene but did not provide information on the copy number and insertion site(s). This is critical to evaluate whether the insertional mutation confounded the observation.

  1. The numbers of transferred embryos, surrogates, and pregnant recipients described in line 204-205 are not consistent with table 2. In table 2, 8/14 surrogates become pregnant by Ultra sound examination. 4 reported days of pregnancy. Among these 4, two were sacrificed at day 33 and fetuses were collected for examination, the other 2 pregnant recipients completed the pregnancy and delivered 4 piglets in total. It is also important to know how many fetuses died before birth.

  1. It is not clearly evident to me that vacuoles are forming in the cytoplasm in figure 2B. Higher magnification images should be provided and the vacuoles should be pointed out. Figure 2B seems to be immunochemistry staining and 2C to be the H&E staining. Could this be the cause?

  1. In addition , the actual measure of PIK3C3 levels and the controls should be included, in addition to the normalized relative expression levels in figure 2A.

  1. Quantitative measure of autophagosomes and autophagolysosomes in the TEM is necessary. Representative images in figure 4C only present a very small area, which could be very biased and misleading. It is important to know if the autophagy level decreased or remained relatively steady in the PIK3C3 over the treatment.
  2. It is challenging for me to understand why LC3B level decrease after ESBB-treatment in the PIK3C3 overexpressing cells. Is it because LC3B itself was degraded by the increased autophagy activity itself? How to interpret the inconsistent observation of decreased LC3B levels and increased ATG and genes involved in the autophagy signaling pathway after EBSS treatment in PIK3C3 overexpressing cells? Authors should comment.

  1. Hyperactivity from PIK3C3 may lead to hyperactivation and hyperproliferation of T cells. The author also discussed the observation of increased immune factors in liver tissue. While this is possible, immune infiltration could be directly measured from the H&E staining and the author should provide a quantitative measure to more directly support this claim. In addition, activation of NF-kB is complicated. Hyper activation of PIK3C3 may activate NF-kb through AKT or mTOR pathway in non-immune cells. It should not be used as direct evidence of autoimmune response.
  2. Immediate ULK1 activation is through phosphorylation of ULK1 which releases it from the repressed complex. In addition to the transcriptional upregulation of ULK1, considering the short time of treatment, the authors should also directly measure the p-ULK1 levels after treatment.

Minor comments:

  1. line 89, critical details in 2.2 vector construction are missing, these include 1)exact sequence being synthesized, 2) sequence info of the pA690, 3)primer sequences used in line 94-95 to PCR the “EN” fragment, which was not defined anywhere in the manuscript. 4) line96-97, is the SalI site added through PCR primers in the PCR amplification of EN fragment? A final sequence map of the construct, pPIK-IRES2-AcGFP1-EN, should be provided. In addition, in Line 99, I may suggest the authors use “verified” instead of “detected”. Is this EN fragment the same as the ENT fragment shown in figure 1A?

  1. Random errors, although with low frequency, in DNA synthesis may randomly introduce mutations. The authors should provide a final sequenced map of the pPIK-IRES2-AcGFP1-EN, at least the PIK3C3 CDS region in the final plasmid that was used for transfecting the fetal fibroblast cells.

  1. Representative images in figure 2B-C, 3A, 4C are intuitive but could also be biased. Authors should provide a quantitative measure over a larger area. In addition, since 4 piglets died at different ages and had different PIK3C3 expression levels, which piglets do those images correspond to?

  1. It is very confusing why the authors did SCNT again using the fatal fibroblasts from positive PIK3C3 transgenic fetuses. Authors should comment
  2. Figure 2A, Since all the piglets are derived from the same genetic clone, why the expression levels of PIK3C3 in different piglets are so different? Is the variation associated with age?

Author Response

We appreciate the time and efforts that you dedicated for providing feedback on our manuscript and are grateful for the insightful comments on and valuable improvements to our paper. We revised the manuscript as per reviewer’s suggestion and revisions are highlighted (blue color) within the manuscript. The point-by-point response to the reviewer comments is attached in a pdf file.

Reviewer 2 Report

In the present study, the authors examined the function of PIK3C3 overexpression on liver and other organs in pigs using PIK3C3 transgenic pigs through (SCNT) following transfecting cells with specific vector for the overexpression of PIK3C3. The experiments are well-designed with interesting results. The manuscript is somewhat well-written. The reviewer, however, have some specific comments as below.

  1. If there are any scientific reasons should the authors choose Diannan miniature over other pig breeds please note in the text. Otherwise, please emphasize somewhere in the text that the results in this study might not apply for other pig breeds such as Landrace, Duroc or Large White which are more commonly used in other laboratories.
  2. Abstract, L34-36: this sentence does confuse the readers. Reading this sentence, my understanding is that “Fetal fibroblasts of PIK3C3 fetuses of the 4 surrogates were used as donor cell for SCNT to generate PIK3C3 transgenic pigs again, i.e second generation of PIK3C3 transgenic pigs. And then, the authors examined the consequences of PIK3C3 overexpression second generation of PIK3C3 transgenic pigs.”
  3. About Figure 1.

Figure 1A: mark all the cut sites for Sal I, EcoRV, Hind III and Apal I.

Figure 1B is rather confusing. Lane 1 is definitely not pPIK-IRES2-AcGFP-EN plasmid as the size is just about 5kb. It seems that Lane 1 is something else, and there are only explanations for Lan2-5. Also, it is written “Mark” not just “M” in the legend.

Figure 1C-F. The band of PIK3C3 does not look like 1652bp to me at all. I would think the band size is close to 1.9kb or at least 1.8kb. Really does not look convincing.

Author Response

(The authors gave the same response as above.)

Reviewer 3 Report

Comments and Suggestions for Authors

This study conducted by Jing Wang, et al., demonstrated that PIK3C3 over expression enhances the vacuolar formation in hepatocytes and enhances apoptosis. In this study they constructed PIK3C3 vector and through somatic cell nuclear transfer (SCNT), they produced the transgenic pigs.  They found that transgenic pigs liver have increased LC3B and LAMP-1-positive cells and accumulated LC3II protein was cleared faster in EBSS-induced autophagy of the WT and PIK3C3 transgenic porcine fibroblast cells (PFCs). They concluded that PIK3C3 overexpression promoted autophagy in the liver and associated molecular mechanism related with the activation of ULK1, AMBR1, DRAM1 and MTOR, causing liver damage in pigs. I have few major and minor comments and after addressing those comments, I recommend this manuscript for publication in MDPI journal “life”. 

Comments; 

  1. Include few sentences related to the PIK3C3 in the introductory section in broad spectrum.
  2. The authors stated that NF-κB, TGF-β1, TLR4, 38 TNF-α and IL-6 significantly increased in livers of PIK3C3 transgenic pigs. Why the authors didn’t checked NK cells markers? NK cell activity can be measure via 51Cr-release assays or flow cytometry based assays.
  3. The authors didn’t mention about the effects of over expression of PIK3C3 on embryo development.
  4. Figure 4, the authors should add kDa in the western blot results. Two bands appear in the image, which one is PIK3C3. The band looks like.
  5. Figure 5, the immunofluorescence results shows that the LC3B expression become reduced with the passage of time (0 to 2 h). If the fluorescence intensity is stronger then please provide the histogram and also provide high quality of immunofluorescence images.

Author Response

(The authors gave the same response as above.)

Round 2

Reviewer 1 Report

The authors have significantly improved the manuscript in the current revision and reasonably addressed most of my questions. However, the major flaw in the experimental design has significantly lowered the value of this study. The authors referenced more PIK3C3 knock-down or knock-out studies to demonstrate the essential functional role of PIK3C3. This does not necessarily mean the overexpression of PIK3C3 is responsible for autophagy in human/animal disease conditions. Importantly, the link between the observed liver damage and PIK3C3 over-expression mediated autophagy is still not convincing based on the current data presented.

Global overexpression of PIK3C3 by multiple folds introduced many confounding factors and made the observed phenotype challenging to interpret. The short-lived nature of those animals made it difficult to be used as a new experimental animal resource for liver disease.

Reviewer 2 Report

The authors have answered all the questions from this reviewer. The reviewer has no more comments or questions.